# Successor Features for Transfer in Reinforcement Learning

**André Barreto, Will Dabney, Rémi Munos, Jonathan J. Hunt,**
**Tom Schaul, Hado van Hasselt, David Silver**

`{andrebarreto,wdabney,munos,jjhunt,schaul,hado,davidsilver}@google.com`

DeepMind

## Abstract

Transfer in reinforcement learning refers to the notion that generalization should occur not only within a task but also across tasks. We propose a transfer framework for the scenario where the reward function changes between tasks but the environment's dynamics remain the same. Our approach rests on two key ideas: *successor features*, a value function representation that decouples the dynamics of the environment from the rewards, and *generalized policy improvement*, a generalization of dynamic programming's policy improvement operation that considers a set of policies rather than a single one. Put together, the two ideas lead to an approach that integrates seamlessly within the reinforcement learning framework and allows the free exchange of information across tasks. The proposed method also provides performance guarantees for the transferred policy even before any learning has taken place. We derive two theorems that set our approach in firm theoretical ground and present experiments that show that it successfully promotes transfer in practice, significantly outperforming alternative methods in a sequence of navigation tasks and in the control of a simulated robotic arm.

## 1 Introduction

Reinforcement learning (RL) provides a framework for the development of situated agents that learn how to behave while interacting with the environment [21]. The basic RL loop is defined in an abstract way so as to capture only the essential aspects of this interaction: an agent receives observations and selects actions to maximize a reward signal. This setup is generic enough to describe tasks of different levels of complexity that may unroll at distinct time scales. For example, in the task of driving a car, an action can be to turn the wheel, make a right turn, or drive to a given location.

Clearly, from the point of view of the designer, it is desirable to describe a task at the highest level of abstraction possible. However, by doing so one may overlook behavioral patterns and inadvertently make the task more difficult than it really is. The task of driving to a location clearly encompasses the subtask of making a right turn, which in turn encompasses the action of turning the wheel. In learning how to drive an agent should be able to identify and exploit such interdependencies. More generally, the agent should be able to break a task into smaller subtasks and use knowledge accumulated in any subset of those to speed up learning in related tasks. This process of leveraging knowledge acquired in one task to improve performance on other tasks is called *transfer* [25, 11].

In this paper we look at one specific type of transfer, namely, when subtasks correspond to different reward functions defined in the same environment. This setup is flexible enough to allow transfer to happen at different levels. In particular, by appropriately defining the rewards one can induce different task decompositions. For instance, the type of hierarchical decomposition involved in the driving example above can be induced by changing the frequency at which rewards are delivered:

a positive reinforcement can be given after each maneuver that is well executed or only at the final destination. Obviously, one can also decompose a task into subtasks that are independent of each other or whose dependency is strictly temporal (that is, when tasks must be executed in a certain order but no single task is clearly "contained" within another).

The types of task decomposition discussed above potentially allow the agent to tackle more complex problems than would be possible were the tasks modeled as a single monolithic challenge. However, in order to apply this divide-and-conquer strategy to its full extent the agent should have an explicit mechanism to promote transfer between tasks. Ideally, we want a transfer approach to have two important properties. First, the flow of information between tasks should not be dictated by a rigid diagram that reflects the relationship between the tasks themselves, such as hierarchical or temporal dependencies. On the contrary, information should be exchanged across tasks whenever useful. Second, rather than being posed as a separate problem, transfer should be integrated into the RL framework as much as possible, preferably in a way that is almost transparent to the agent.

In this paper we propose an approach for transfer that has the two properties above. Our method builds on two conceptual pillars that complement each other. The first is a generalization of Dayan's [7] *successor representation*. As the name suggests, in this representation scheme each state is described by a prediction about the future occurrence of all states under a fixed policy. We present a generalization of Dayan's idea which extends the original scheme to continuous spaces and also facilitates the use of approximation. We call the resulting scheme *successor features*. As will be shown, successor features lead to a representation of the value function that naturally decouples the dynamics of the environment from the rewards, which makes them particularly suitable for transfer.

The second pillar of our framework is a generalization of Bellman's [3] classic policy improvement theorem that extends the original result from one to multiple decision policies. This novel result shows how knowledge about a set of tasks can be transferred to a new task in a way that is completely integrated within RL. It also provides performance guarantees on the new task *before* any learning has taken place, which opens up the possibility of constructing a library of "skills" that can be reused to solve previously unseen tasks. In addition, we present a theorem that formalizes the notion that an agent should be able to perform well on a task if it has seen a similar task before—something clearly desirable in the context of transfer. Combined, the two results above not only set our approach in firm ground but also outline the mechanics of how to actually implement transfer. We build on this knowledge to propose a concrete method and evaluate it in two environments, one encompassing a sequence of navigation tasks and the other involving the control of a simulated two-joint robotic arm.

## 2 Background and problem formulation

As usual, we assume that the interaction between agent and environment can be modeled as a *Markov decision process* (MDP, Puterman, [19]). An MDP is defined as a tuple $M \equiv (\mathcal{S}, \mathcal{A}, p, R, \gamma)$. The sets $\mathcal{S}$ and $\mathcal{A}$ are the state and action spaces, respectively; here we assume that $\mathcal{S}$ and $\mathcal{A}$ are finite whenever such an assumption facilitates the presentation, but most of the ideas readily extend to continuous spaces. For each $s \in \mathcal{S}$ and $a \in \mathcal{A}$ the function $p(\cdot|s,a)$ gives the next-state distribution upon taking action $a$ in state $s$. We will often refer to $p(\cdot|s,a)$ as the *dynamics* of the MDP. The reward received at transition $s \xrightarrow{a} s'$ is given by the random variable $R(s,a,s')$; usually one is interested in the expected value of this variable, which we will denote by $r(s,a,s')$ or by $r(s,a) = \mathrm{E}_{S' \sim p(\cdot|s,a)}[r(s,a,S')]$. The discount factor $\gamma \in [0,1)$ gives smaller weights to future rewards.

The objective of the agent in RL is to find a policy $\pi$—a mapping from states to actions—that maximizes the expected discounted sum of rewards, also called the *return* $G_t = \sum_{i=0}^{\infty} \gamma^i R_{t+i+1}$, where $R_t = R(S_t, A_t, S_{t+1})$. One way to address this problem is to use methods derived from *dynamic programming* (DP), which heavily rely on the concept of a *value function* [19]. The *action-value function* of a policy $\pi$ is defined as

$$Q^\pi(s,a) \equiv \mathrm{E}^\pi \left[ G_t \,|\, S_t = s, A_t = a \right], \tag{1}$$

where $\mathrm{E}^\pi[\cdot]$ denotes expected value when following policy $\pi$. Once the action-value function of a particular policy $\pi$ is known, we can derive a new policy $\pi'$ which is *greedy* with respect to $Q^\pi(s,a)$, that is, $\pi'(s) \in \mathrm{argmax}_a Q^\pi(s,a)$. Policy $\pi'$ is guaranteed to be at least as good as (if not better than) policy $\pi$. The computation of $Q^\pi(s,a)$ and $\pi'$, called *policy evaluation* and *policy improvement*, define the basic mechanics of RL algorithms based on DP; under certain conditions their successive application leads to an optimal policy $\pi^*$ that maximizes the expected return from every $s \in \mathcal{S}$ [21].

In this paper we are interested in the problem of *transfer*, which we define as follows. Let $\mathcal{T}, \mathcal{T}'$ be two sets of tasks such that $\mathcal{T}' \subset \mathcal{T}$, and let $t$ be any task. Then there is *transfer* if, after training on $\mathcal{T}$, the agent always performs as well or better on task $t$ than if only trained on $\mathcal{T}'$. Note that $\mathcal{T}'$ can be the empty set. In this paper a task will be defined as a specific instantiation of the reward function $R(s, a, s')$ for a given MDP. In Section 4 we will revisit this definition and make it more formal.

## 3   Successor features

In this section we present the concept that will serve as a cornerstone for the rest of the paper. We start by presenting a simple reward model and then show how it naturally leads to a generalization of Dayan's [7] successor representation (SR).

Suppose that the expected one-step reward associated with transition $(s, a, s')$ can be computed as

$$r(s, a, s') = \phi(s, a, s')^\top \mathbf{w}, \tag{2}$$

where $\phi(s, a, s') \in \mathbb{R}^d$ are features of $(s, a, s')$ and $\mathbf{w} \in \mathbb{R}^d$ are weights. Expression (2) is not restrictive because we are not making any assumptions about $\phi(s, a, s')$: if we have $\phi_i(s, a, s') = r(s, a, s')$ for some $i$, for example, we can clearly recover any reward function exactly. To simplify the notation, let $\phi_t = \phi(s_t, a_t, s_{t+1})$. Then, by simply rewriting the definition of the action-value function in (1) we have

$$
\begin{aligned}
Q^\pi(s, a) &= \mathrm{E}^\pi \left[ r_{t+1} + \gamma r_{t+2} + \dots \mid S_t = s, A_t = a \right] \\
&= \mathrm{E}^\pi \left[ \phi_{t+1}^\top \mathbf{w} + \gamma \phi_{t+2}^\top \mathbf{w} + \dots \mid S_t = s, A_t = a \right] \\
&= \mathrm{E}^\pi \left[ \sum_{i=t}^\infty \gamma^{i-t} \phi_{i+1} \mid S_t = s, A_t = a \right]^\top \mathbf{w} = \psi^\pi(s, a)^\top \mathbf{w}.
\end{aligned}
\tag{3}
$$

The decomposition (3) has appeared before in the literature under different names and interpretations, as discussed in Section 6. Since here we propose to look at (3) as an extension of Dayan's [7] SR, we call $\psi^\pi(s, a)$ the *successor features* (SFs) of $(s, a)$ under policy $\pi$.

The $i^{\text{th}}$ component of $\psi^\pi(s, a)$ gives the expected discounted sum of $\phi_i$ when following policy $\pi$ starting from $(s, a)$. In the particular case where $\mathcal{S}$ and $\mathcal{A}$ are finite and $\phi$ is a tabular representation of $\mathcal{S} \times \mathcal{A} \times \mathcal{S}$—that is, $\phi(s, a, s')$ is a one-hot vector in $\mathbb{R}^{|\mathcal{S}|^2|\mathcal{A}|}$—$\psi^\pi(s, a)$ is the discounted sum of occurrences, under $\pi$, of each possible transition. This is essentially the concept of SR extended from the space $\mathcal{S}$ to the set $\mathcal{S} \times \mathcal{A} \times \mathcal{S}$ [7].

One of the contributions of this paper is precisely to generalize SR to be used with function approximation, but the exercise of deriving the concept as above provides insights already in the tabular case. To see this, note that in the tabular case the entries of $\mathbf{w} \in \mathbb{R}^{|\mathcal{S}|^2|\mathcal{A}|}$ are the function $r(s, a, s')$ and suppose that $r(s, a, s') \neq 0$ in only a small subset $\mathcal{W} \subset \mathcal{S} \times \mathcal{A} \times \mathcal{S}$. From (2) and (3), it is clear that the cardinality of $\mathcal{W}$, and not of $\mathcal{S} \times \mathcal{A} \times \mathcal{S}$, is what effectively defines the dimension of the representation $\psi^\pi$, since there is no point in having $d > |\mathcal{W}|$. Although this fact is hinted at by Dayan [7], it becomes more apparent when we look at SR as a particular case of SFs.

SFs extend SR in two other ways. First, the concept readily applies to continuous state and action spaces without any modification. Second, by explicitly casting (2) and (3) as inner products involving feature vectors, SFs make it evident how to incorporate function approximation: as will be shown, these vectors can be learned from data.

The representation in (3) requires two components to be learned, $\mathbf{w}$ and $\psi^\pi$. Since the latter is the expected discounted sum of $\phi$ under $\pi$, we must either be given $\phi$ or learn it as well. Note that approximating $r(s, a, s') \approx \phi(s, a, s')^\top \tilde{\mathbf{w}}$ is a supervised learning problem, so we can use well-understood techniques from the field to learn $\tilde{\mathbf{w}}$ (and potentially $\tilde{\phi}$, too) [9]. As for $\psi^\pi$, we note that

$$\psi^\pi(s, a) = \phi_{t+1} + \gamma E^\pi [\psi^\pi(S_{t+1}, \pi(S_{t+1})) \mid S_t = s, A_t = a], \tag{4}$$

that is, SFs satisfy a Bellman equation in which $\phi_i$ play the role of rewards—something also noted by Dayan [7] regarding SR. Therefore, in principle *any* RL method can be used to compute $\psi^\pi$ [24].

The SFs $\psi^\pi$ summarize the dynamics induced by $\pi$ in a given environment. As shown in (3), this allows for a modular representation of $Q^\pi$ in which the MDP's dynamics are decoupled from its

rewards, which are captured by the weights $\mathbf{w}$. One potential benefit of having such a decoupled representation is that only the relevant module must be relearned when either the dynamics or the reward changes, which may serve as an argument in favor of adopting SFs as a general approximation scheme for RL. However, in this paper we focus on a scenario where the decoupled value-function approximation provided by SFs is exploited to its full extent, as we discuss next.

## 4 Transfer via successor features

We now return to the discussion about transfer in RL. As described, we are interested in the scenario where all components of an MDP are fixed, except for the reward function. One way to formalize this model is through (2): if we suppose that $\phi \in \mathbb{R}^d$ is fixed, any $\mathbf{w} \in \mathbb{R}^d$ gives rise to a new MDP. Based on this observation, we define

$$\mathcal{M}^{\phi}(\mathcal{S}, \mathcal{A}, p, \gamma) \equiv \{M(\mathcal{S}, \mathcal{A}, p, r, \gamma) \mid r(s, a, s') = \phi(s, a, s')^{\top} \mathbf{w}\}, \tag{5}$$

that is, $\mathcal{M}^{\phi}$ is the set of MDPs induced by $\phi$ through all possible instantiations of $\mathbf{w}$. Since what differentiates the MDPs in $\mathcal{M}^{\phi}$ is essentially the agent's goal, we will refer to $M_i \in \mathcal{M}^{\phi}$ as a *task*. The assumption is that we are interested in solving (a subset of) the tasks in the environment $\mathcal{M}^{\phi}$.

Definition (5) is a natural way of modeling some scenarios of interest. Think, for example, how the desirability of water or food changes depending on whether an animal is thirsty or hungry. One way to model this type of preference shifting is to suppose that the vector $\mathbf{w}$ appearing in (2) reflects the taste of the agent at any given point in time [17]. Further in the paper we will present experiments that reflect this scenario. For another illustrative example, imagine that the agent's goal is to produce and sell a combination of goods whose production line is relatively stable but whose prices vary considerably over time. In this case updating the price of the products corresponds to picking a new $\mathbf{w}$. A slightly different way of motivating (5) is to suppose that the environment itself is changing, that is, the element $\mathbf{w}_i$ indicates not only desirability, but also availability, of feature $\phi_i$.

In the examples above it is desirable for the agent to build on previous experience to improve its performance on a new setup. More concretely, if the agent knows good policies for the set of tasks $\mathcal{M} \equiv \{M_1, M_2, ..., M_n\}$, with $M_i \in \mathcal{M}^{\phi}$, it should be able to leverage this knowledge to improve its behavior on a new task $M_{n+1}$—that is, it should perform better than it would had it been exposed to only a subset of the original tasks, $\mathcal{M}' \subset \mathcal{M}$. We can assess the performance of an agent on task $M_{n+1}$ based on the value function of the policy followed after $\mathbf{w}_{n+1}$ has become available but *before* any policy improvement has taken place in $M_{n+1}$.[1] More precisely, suppose that an agent has been exposed to each one of the tasks $M_i \in \mathcal{M}'$. Based on this experience, and on the new $\mathbf{w}_{n+1}$, the agent computes a policy $\pi'$ that will define its initial behavior in $M_{n+1}$. Now, if we repeat the experience replacing $\mathcal{M}'$ with $\mathcal{M}$, the resulting policy $\pi$ should be such that $Q^{\pi}(s, a) \geq Q^{\pi'}(s, a)$ for all $(s, a) \in \mathcal{S} \times \mathcal{A}$.

Now that our setup is clear we can start to describe our solution for the transfer problem discussed above. We do so in two stages. First, we present a generalization of DP's notion of policy improvement whose interest may go beyond the current work. We then show how SFs can be used to implement this generalized form of policy improvement in an efficient and elegant way.

### 4.1 Generalized policy improvement

One of the fundamental results in RL is Bellman's [3] *policy improvement theorem*. In essence, the theorem states that acting greedily with respect to a policy's value function gives rise to another policy whose performance is no worse than the former's. This is the driving force behind DP, and most RL algorithms that compute a value function are exploiting Bellman's result in one way or another.

In this section we extend the policy improvement theorem to the scenario where the new policy is to be computed based on the value functions of a *set* of policies. We show that this extension can be done in a natural way, by acting greedily with respect to the maximum over the value functions available. Our result is summarized in the theorem below.

**Theorem 1. (Generalized Policy Improvement)** *Let $\pi_1, \pi_2, ..., \pi_n$ be $n$ decision policies and let $\tilde{Q}^{\pi_1}, \tilde{Q}^{\pi_2}, ..., \tilde{Q}^{\pi_n}$ be approximations of their respective action-value functions such that*

$$|Q^{\pi_i}(s,a) - \tilde{Q}^{\pi_i}(s,a)| \leq \epsilon \text{ for all } s \in \mathcal{S}, a \in \mathcal{A}, \text{ and } i \in \{1,2,...,n\}. \tag{6}$$

*Define*

$$\pi(s) \in \operatorname*{argmax}_a \max_i \tilde{Q}^{\pi_i}(s,a). \tag{7}$$

*Then,*

$$Q^\pi(s,a) \geq \max_i Q^{\pi_i}(s,a) - \frac{2}{1-\gamma}\epsilon \tag{8}$$

*for any $s \in \mathcal{S}$ and $a \in \mathcal{A}$, where $Q^\pi$ is the action-value function of $\pi$.*

The proofs of our theoretical results are in the supplementary material. As one can see, our theorem covers the case where the policies' value functions are not computed exactly, either because function approximation is used or because some exact algorithm has not be run to completion. This error is captured by $\epsilon$ in (6), which re-appears as a penalty term in the lower bound (8). Such a penalty is inherent to the presence of approximation in RL, and in fact it is identical to the penalty incurred in the single-policy case (see *e.g.* Bertsekas and Tsitsiklis's Proposition 6.1 [5]).

In order to contextualize generalized policy improvement (GPI) within the broader scenario of DP, suppose for a moment that $\epsilon = 0$. In this case Theorem 1 states that $\pi$ will perform no worse than *all* of the policies $\pi_1, \pi_2, ..., \pi_n$. This is interesting because in general $\max_i Q^{\pi_i}$—the function used to induce $\pi$—is not the value function of any particular policy. It is not difficult to see that $\pi$ will be strictly better than all previous policies if no single policy dominates all other policies, that is, if $\operatorname{argmax}_i \max_a \tilde{Q}^{\pi_i}(s,a) \cap \operatorname{argmax}_i \max_a \tilde{Q}^{\pi_i}(s',a) = \emptyset$ for some $s, s' \in \mathcal{S}$. If one policy does dominate all others, GPI reduces to the original policy improvement theorem.

If we consider the usual DP loop, in which policies of increasing performance are computed in sequence, our result is not of much use because the most recent policy will always dominate all others. Another way of putting it is to say that after Theorem 1 is applied once adding the resulting $\pi$ to the set $\{\pi_1, \pi_2, ..., \pi_n\}$ will reduce the next improvement step to standard policy improvement, and thus the policies $\pi_1, \pi_2, ..., \pi_n$ can be simply discarded. There are however two situations in which our result may be of interest. One is when we have many policies $\pi_i$ being evaluated in parallel. In this case GPI provides a principled strategy for combining these policies. The other situation in which our result may be useful is when the underlying MDP changes, as we discuss next.

### 4.2 Generalized policy improvement with successor features

We start this section by extending our notation slightly to make it easier to refer to the quantities involved in transfer learning. Let $M_i$ be a task in $\mathcal{M}^\phi$ defined by $\mathbf{w}_i \in \mathbb{R}^d$. We will use $\pi_i^*$ to refer to an optimal policy of MDP $M_i$ and use $Q_i^{\pi_i^*}$ to refer to its value function. The value function of $\pi_i^*$ when executed in $M_j \in \mathcal{M}^\phi$ will be denoted by $Q_j^{\pi_i^*}$.

Suppose now that an agent has computed optimal policies for the tasks $M_1, M_2, ..., M_n \in \mathcal{M}^\phi$. Suppose further that when presented with a new task $M_{n+1}$ the agent computes $\{Q_{n+1}^{\pi_1^*}, Q_{n+1}^{\pi_2^*}, ..., Q_{n+1}^{\pi_n^*}\}$, the evaluation of each $\pi_i^*$ under the new reward function induced by $\mathbf{w}_{n+1}$. In this case, applying the GPI theorem to the newly-computed set of value functions will give rise to a policy that performs at least as well as a policy based on any subset of these, including the empty set. Thus, this strategy satisfies our definition of successful transfer.

There is a caveat, though. Why would one waste time computing the value functions of $\pi_1^*, \pi_2^*, ..., \pi_n^*$, whose performance in $M_{n+1}$ may be mediocre, if the same amount of resources can be allocated to compute a sequence of $n$ policies with increasing performance? This is where SFs come into play. Suppose that we have learned the functions $Q_i^{\pi_i^*}$ using the representation scheme shown in (3). Now, if the reward changes to $r_{n+1}(s,a,s') = \phi(s,a,s')^\top \mathbf{w}_{n+1}$, as long as we have $\mathbf{w}_{n+1}$ we can compute the new value function of $\pi_i^*$ by simply making $Q_{n+1}^{\pi_i^*}(s,a) = \boldsymbol{\psi}^{\pi_i^*}(s,a)^\top \mathbf{w}_{n+1}$. This reduces the computation of all $Q_{n+1}^{\pi_i^*}$ to the much simpler supervised problem of approximating $\mathbf{w}_{n+1}$.

Once the functions $Q_{n+1}^{\pi_i^*}$ have been computed, we can apply GPI to derive a policy $\pi$ whose performance on $M_{n+1}$ is no worse than the performance of $\pi_1^*, \pi_2^*, ..., \pi_n^*$ on the same task. A

question that arises in this case is whether we can provide stronger guarantees on the performance of $\pi$ by exploiting the structure shared by the tasks in $\mathcal{M}^\phi$. The following theorem answers this question in the affirmative.

**Theorem 2.** *Let $M_i \in \mathcal{M}^\phi$ and let $Q_i^{\pi_j^*}$ be the action-value function of an optimal policy of $M_j \in \mathcal{M}^\phi$ when executed in $M_i$. Given approximations $\{\tilde{Q}_i^{\pi_1^*}, \tilde{Q}_i^{\pi_2^*}, ..., \tilde{Q}_i^{\pi_n^*}\}$ such that*

$$\left| Q_i^{\pi_j^*}(s,a) - \tilde{Q}_i^{\pi_j^*}(s,a) \right| \le \epsilon \tag{9}$$

*for all $s \in \mathcal{S}$, $a \in \mathcal{A}$, and $j \in \{1, 2, ..., n\}$, let $\pi(s) \in \operatorname{argmax}_a \max_j \tilde{Q}_i^{\pi_j^*}(s,a)$. Finally, let $\phi_{\max} = \max_{s,a} ||\phi(s,a)||$, where $|| \cdot ||$ is the norm induced by the inner product adopted. Then,*

$$Q_i^{\pi_i^*}(s,a) - Q_i^\pi(s,a) \le \frac{2}{1-\gamma} \left( \phi_{\max} \min_j ||\mathbf{w}_i - \mathbf{w}_j|| + \epsilon \right). \tag{10}$$

Note that we used $M_i$ rather than $M_{n+1}$ in the theorem's statement to remove any suggestion of order among the tasks. Theorem 2 is a specialization of Theorem 1 for the case where the set of value functions used to compute $\pi$ are associated with tasks in the form of (5). As such, it provides stronger guarantees: instead of comparing the performance of $\pi$ with that of the previously-computed policies $\pi_j$, Theorem 2 quantifies the loss incurred by following $\pi$ as opposed to one of $M_i$'s optimal policies.

As shown in (10), the loss $Q_i^{\pi_i^*}(s,a) - Q_i^\pi(s,a)$ is upper-bounded by two terms. The term $2\phi_{\max}\min_j||\mathbf{w}_i - \mathbf{w}_j||/(1-\gamma)$ is of more interest here because it reflects the structure of $\mathcal{M}^\phi$. This term is a multiple of the distance between $\mathbf{w}_i$, the vector describing the task we are currently interested in, and the closest $\mathbf{w}_j$ for which we have computed a policy. This formalizes the intuition that the agent should perform well in task $\mathbf{w}_i$ if it has solved a similar task before. More generally, the term in question relates the concept of distance in $\mathbb{R}^d$ with difference in performance in $\mathcal{M}^\phi$. Note that this correspondence depends on the specific set of features $\phi$ used, which raises the interesting question of how to define $\phi$ such that tasks that are close in $\mathbb{R}^d$ induce policies that are also similar in some sense. Regardless of how exactly $\phi$ is defined, the bound (10) allows for powerful extrapolations. For example, by covering the relevant subspace of $\mathbb{R}^d$ with balls of appropriate radii centered at $\mathbf{w}_j$ we can provide performance guarantees for *any* task $\mathbf{w}$ [14]. This corresponds to building a library of *options* (or "skills") that can be used to solve any task in a (possibly infinite) set [22]. In Section 5 we illustrate this concept with experiments.

Although Theorem 2 is inexorably related to the characterization of $\mathcal{M}^\phi$ in (5), it does not depend on the definition of SFs in any way. Here SFs are the *mechanism* used to efficiently apply the protocol suggested by Theorem 2. When SFs are used the value function approximations are given by $\tilde{Q}_i^{\pi_j^*}(s,a) = \tilde{\psi}^{\pi_j^*}(s,a)^\top \tilde{\mathbf{w}}_i$. The modules $\tilde{\psi}^{\pi_j^*}$ are computed and stored when the agent is learning the tasks $M_j$; when faced with a new task $M_i$ the agent computes an approximation of $\mathbf{w}_i$, which is a supervised learning problem, and then uses the GPI policy $\pi$ defined in Theorem 2 to learn $\tilde{\psi}^{\pi_i^*}$. Note that we do not assume that either $\psi^{\pi_j^*}$ or $\mathbf{w}_i$ is computed exactly: the effect of errors in $\tilde{\psi}^{\pi_j^*}$ and $\tilde{\mathbf{w}}_i$ are accounted for by the term $\epsilon$ appearing in (9). As shown in (10), if $\epsilon$ is small and the agent has seen enough tasks the performance of $\pi$ on $M_i$ should already be good, which suggests it may also speed up the process of learning $\tilde{\psi}^{\pi_i^*}$.

Interestingly, Theorem 2 also provides guidance for some practical algorithmic choices. Since in an actual implementation one wants to limit the number of SFs $\tilde{\psi}^{\pi_j^*}$ stored in memory, the corresponding vectors $\tilde{\mathbf{w}}_j$ can be used to decide which ones to keep. For example, one can create a new $\tilde{\psi}^{\pi_i^*}$ only when $\min_j||\tilde{\mathbf{w}}_i - \tilde{\mathbf{w}}_j||$ is above a given threshold; alternatively, once the maximum number of SFs has been reached, one can replace $\tilde{\psi}^{\pi_k^*}$, where $k = \operatorname{argmin}_j||\tilde{\mathbf{w}}_i - \tilde{\mathbf{w}}_j||$ (here $\mathbf{w}_i$ is the current task).

## 5 Experiments

In this section we present our main experimental results. Additional details, along with further results and analysis, can be found in Appendix B of the supplementary material.

The first environment we consider involves navigation tasks defined over a two-dimensional continuous space composed of four rooms (Figure 1). The agent starts in one of the rooms and must reach a

goal region located in the farthest room. The environment has objects that can be picked up by the agent by passing over them. Each object belongs to one of three classes determining the associated reward. The objective of the agent is to pick up the "good" objects and navigate to the goal while avoiding "bad" objects. The rewards associated with object classes change at every 20 000 transitions, giving rise to very different tasks (Figure 1). The goal is to maximize the sum of rewards accumulated over a sequence of 250 tasks, with each task's rewards sampled uniformly from $[-1, 1]^3$.

We defined a straightforward instantiation of our approach in which both $\tilde{\mathbf{w}}$ and $\tilde{\psi}^\pi$ are computed incrementally in order to minimize losses induced by (2) and (4). Every time the task changes the current $\tilde{\psi}^{\pi_i}$ is stored and a new $\tilde{\psi}^{\pi_{i+1}}$ is created. We call this method SFQL as a reference to the fact that SFs are learned through an algorithm analogous to $Q$-learning (QL)—which is used as a baseline in our comparisons [27]. As a more challenging reference point we report results for a transfer method called *probabilistic policy reuse* [8]. We adopt a version of the algorithm that builds on

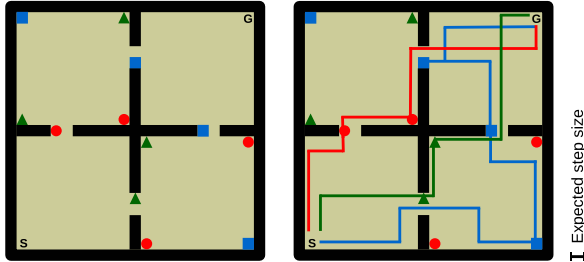

Figure 1: Environment layout and some examples of optimal trajectories associated with specific tasks. The shapes of the objects represent their classes; 'S' is the start state and 'G' is the goal.

QL and reuses all policies learned. The resulting method, PRQL, is thus directly comparable to SFQL. The details of QL, PRQL, and SFQL, including their pseudo-codes, are given in Appendix B.

We compared two versions of SFQL. In the first one, called SFQL-$\phi$, we assume the agent has access to features $\phi$ that perfectly predict the rewards, as in (2). The second version of our agent had to learn an approximation $\tilde{\phi} \in \mathbb{R}^h$ directly from data collected by QL in the first 20 tasks. Note that $h$ may not coincide with the true dimension of $\phi$, which in this case is 4; we refer to the different instances of our algorithm as SFQL-$h$. The process of learning $\tilde{\phi}$ followed the multi-task learning protocol proposed by Caruana [6] and Baxter [2], and described in detail in Appendix B.

The results of our experiments can be seen in Figure 2. As shown, all versions of SFQL significantly outperform the other two methods, with an improvement on the average return of more than $100\%$ when compared to PRQL, which itself improves on QL by around $100\%$. Interestingly, SFQL-$h$ seems to achieve good overall performance *faster* than SFQL-$\phi$, even though the latter uses features that allow for an exact representation of the rewards. One possible explanation is that, unlike their counterparts $\phi_i$, the features $\tilde{\phi}_i$ are activated over most of the space $\mathcal{S} \times \mathcal{A} \times \mathcal{S}$, which results in a dense pseudo-reward signal that facilitates learning.

The second environment we consider is a set of control tasks defined in the MuJoCo physics engine [26]. Each task consists in moving a two-joint torque-controlled simulated robotic arm to a

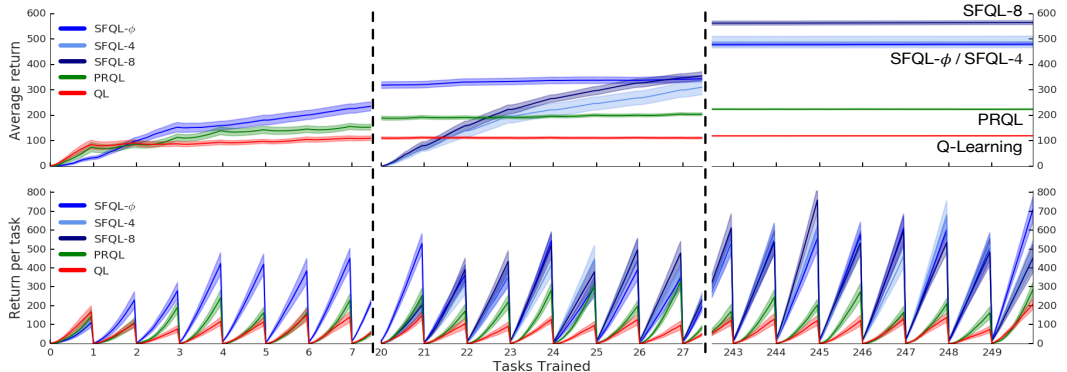

Figure 2: Average and cumulative return per task in the four-room domain. SFQL-$h$ receives no reward during the first 20 tasks while learning $\tilde{\phi}$. Error-bands show one standard error over 30 runs.

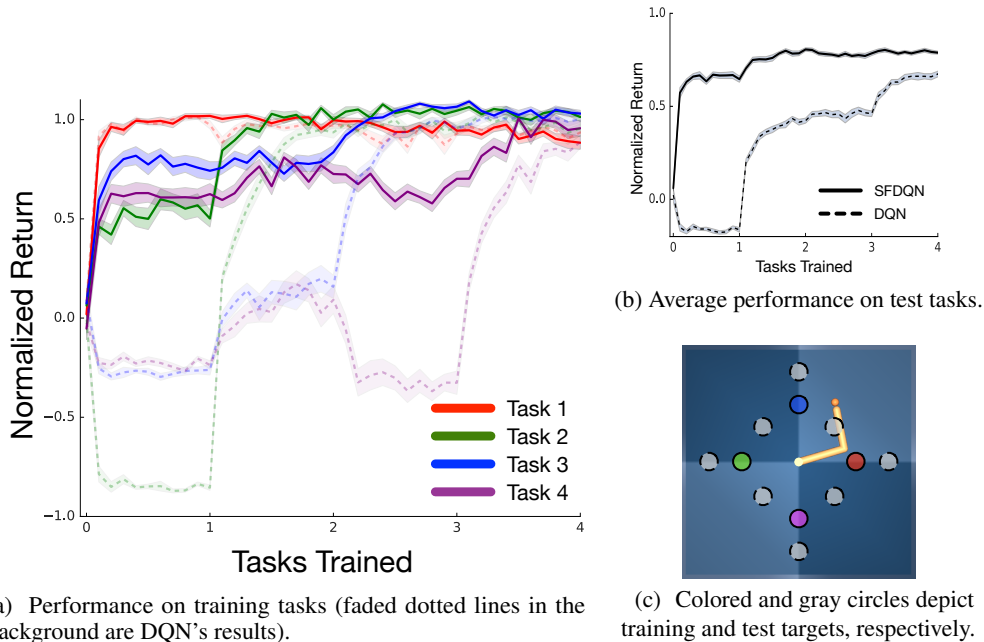

(a) Performance on training tasks (faded dotted lines in the background are DQN's results).

(b) Average performance on test tasks.

(c) Colored and gray circles depict training and test targets, respectively.

Figure 3: Normalized return on the reacher domain: '1' corresponds to the average result achieved by DQN after learning each task separately and '0' corresponds to the average performance of a randomly-initialized agent (see Appendix B for details). SFDQN's results were obtained using the GPI policies $\pi_i(s)$ defined in the text. Shading shows one standard error over 30 runs.

specific target location; thus, we refer to this environment as "the reacher domain." We defined 12 tasks, but only allowed the agents to train in 4 of them (Figure 3c). This means that the agent must be able to perform well on tasks that it has never experienced during training.

In order to solve this problem, we adopted essentially the same algorithm as above, but we replaced QL with Mnih et al.'s DQN—both as a baseline and as the basic engine underlying the SF agent [15]. The resulting method, which we call SFDQN, is an illustration of how our method can be naturally combined with complex nonlinear approximators such as neural networks. The features $\phi_i$ used by SFDQN are the negation of the distances to the center of the 12 target regions. As usual in experiments of this type, we give the agents a description of the current task: for DQN the target coordinates are given as inputs, while for SFDQN this is provided as an one-hot vector $\mathbf{w}_t \in \mathbb{R}^{12}$ [12]. Unlike in the previous experiment, in the current setup each transition was used to train all four $\tilde{\psi}^{\pi_i}$ through losses derived from (4). Here $\pi_i$ is the GPI policy on the $\mathbf{i}^{\text{th}}$ task: $\pi_i(s) \in \operatorname{argmax}_a \max_j \tilde{\psi}_j(s, a)^\top \mathbf{w}_i$.

Results are shown in Figures 3a and 3b. Looking at the training curves, we see that whenever a task is selected for training SFDQN's return on that task quickly improves and saturates at near-optimal performance. The interesting point to be noted is that, when learning a given task, SFDQN's performance also improves in all other tasks, including the test ones, for which it does not have specialized policies. This illustrates how the combination of SFs and GPI can give rise to flexible agents able to perform well in *any* task of a set of tasks with shared dynamics—which in turn can be seen as both a form of temporal abstraction and a step towards more general hierarchical RL [22, 1].

# 6 Related work

Mehta et al.'s [14] approach for transfer learning is probably the closest work to ours in the literature. There are important differences, though. First, Mehta et al. [14] assume that both $\phi$ and $\mathbf{w}$ are always observable quantities provided by the environment. They also focus on average reward RL, in which the quality of a decision policy can be characterized by a single scalar. This reduces the process of selecting a policy for a task to one decision made at the outset, which is in clear contrast with GPI.

The literature on transfer learning has other methods that relate to ours [25, 11]. Among the algorithms designed for the scenario considered here, two approaches are particularly relevant because they also reuse old policies. One is Fernández et al.'s [8] probabilistic policy reuse, adopted in our experiments and described in Appendix B. The other approach, by Bernstein [4], corresponds to using our method but relearning all $\tilde{\boldsymbol{\psi}}^{\pi_i}$ from scratch at each new task.

When we look at SFs strictly as a representation scheme, there are clear similarities with Littman et al.'s [13] predictive state representation (PSR). Unlike SFs, though, PSR tries to summarize the dynamics of the entire environment rather than of a single policy $\pi$. A scheme that is perhaps closer to SFs is the value function representation sometimes adopted in inverse RL [18].

SFs are also related to Sutton et al.'s [23] *general value functions* (GVFs), which extend the notion of value function to also include "pseudo-rewards." If we see $\phi_i$ as a pseudo-reward, $\psi_i^\pi(s, a)$ becomes a particular case of GVF. Beyond the technical similarities, the connection between SFs and GVFs uncovers some principles underlying both lines of work that, when contrasted, may benefit both. On one hand, Sutton et al.'s [23] and Modayil et al.'s [16] hypothesis that relevant knowledge about the world can be expressed in the form of many predictions naturally translates to SFs: if $\phi$ is expressive enough, the agent should be able to represent *any* relevant reward function. Conversely, SFs not only provide a concrete way of using this knowledge, they also suggest a possible criterion to select the pseudo-rewards $\phi_i$, since ultimately we are only interested in features that help in the approximation $\phi(s, a, s')^\top \tilde{\mathbf{w}} \approx r(s, a, s')$.

Another generalization of value functions that is related to SFs is Schaul et al.'s [20] *universal value function approximators* (UVFAs). UVFAs extend the notion of value function to also include as an argument an abstract representation of a "goal," which makes them particularly suitable for transfer. The function $\max_j \tilde{\boldsymbol{\psi}}^{\pi_j^*}(s, a)^\top \tilde{\mathbf{w}}$ used in our framework can be seen as a function of $s$, $a$, and $\tilde{\mathbf{w}}$—the latter a generic way of representing a goal—, and thus in some sense this representation *is* a UVFA. The connection between SFs and UVFAs raises an interesting point: since under this interpretation $\tilde{\mathbf{w}}$ is simply the description of a task, it can in principle be a direct function of the observations, which opens up the possibility of the agent determining $\tilde{\mathbf{w}}$ even *before* seeing any rewards.

As discussed, our approach is also related to temporal abstraction and hierarchical RL: if we look at $\boldsymbol{\psi}^\pi$ as instances of Sutton et al.'s [22] *options*, acting greedily with respect to the maximum over their value functions corresponds in some sense to planning at a higher level of temporal abstraction (that is, each $\boldsymbol{\psi}^\pi(s, a)$ is associated with an option that terminates after a single step). This is the view adopted by Yao et al. [28], whose *universal option model* closely resembles our approach in some aspects (the main difference being that they do not do GPI).

Finally, there have been previous attempts to combine SR and neural networks. Kulkarni et al. [10] and Zhang et al. [29] propose similar architectures to jointly learn $\tilde{\boldsymbol{\psi}}^\pi(s, a)$, $\tilde{\boldsymbol{\phi}}(s, a, s')$ and $\tilde{\mathbf{w}}$. Although neither work exploits SFs for GPI, they both discuss other uses of SFs for transfer. In principle the proposed (or similar) architectures can also be used within our framework.

# 7 Conclusion

This paper builds on two concepts, both of which are generalizations of previous ideas. The first one is SFs, a generalization of Dayan's [7] SR that extends the original definition from discrete to continuous spaces and also facilitates the use of function approximation. The second concept is GPI, formalized in Theorem 1. As the name suggests, this result extends Bellman's [3] classic policy improvement theorem from a single to multiple policies.

Although SFs and GPI are of interest on their own, in this paper we focus on their combination to induce transfer. The resulting framework is an elegant extension of DP's basic setting that provides a solid foundation for transfer in RL. As a complement to the proposed transfer approach, we derived a theoretical result, Theorem 2, that formalizes the intuition that an agent should perform well on a novel task if it has seen a similar task before. We also illustrated with a comprehensive set of experiments how the combination of SFs and GPI promotes transfer in practice.

We believe the proposed ideas lay out a general framework for transfer in RL. By specializing the basic components presented one can build on our results to derive agents able to perform well across a wide variety of tasks, and thus extend the range of environments that can be successfully tackled.

## Acknowledgments

The authors would like to thank Joseph Modayil for the invaluable discussions during the development of the ideas described in this paper. We also thank Peter Dayan, Matt Botvinick, Marc Bellemare, and Guy Lever for the excellent comments, and Dan Horgan and Alexander Pritzel for their help with the experiments. Finally, we thank the anonymous reviewers for their comments and suggestions to improve the paper.

## Footnotes

[1]Of course $\mathbf{w}_{n+1}$ can, and will be, learned, as discussed in Section 4.2 and illustrated in Section 5. Here we assume that $\mathbf{w}_{n+1}$ is given to make our performance criterion clear.

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
