[Supplementary Material]

# Successor Features for
# Transfer in Reinforcement Learning
## Supplementary Material

**André Barreto**, **Will Dabney**, **Rémi Munos**, **Jonathan J. Hunt**,
**Tom Schaul**, **Hado van Hasselt**, **David Silver**

{andrebarreto,wdabney,munos,jjhunt,schaul,hado,davidsilver}@google.com

DeepMind

## Abstract

In this supplement we give details of the theory and experiments that had to be left out of the main paper due to the space limit. For the convenience of the reader the statements of the theoretical results are reproduced before the respective proofs. We also provide a thorough description of the protocol used to carry out our experiments, present details of the algorithms, including their pseudo-code, and report additional empirical analysis that could not be included in the paper. The numbering of sections, equations, and figures resume that used in the main paper, so we refer to these elements as if paper and supplement were a single document. We also cite references listed in the main paper.

## A  Proofs of theoretical results

**Theorem 1.  (Generalized Policy Improvement)** *Let $\pi_1$, $\pi_2$, ..., $\pi_n$ be $n$ decision policies and let $\tilde{Q}^{\pi_1}$, $\tilde{Q}^{\pi_2}$, ..., $\tilde{Q}^{\pi_n}$ be approximations of their respective action-value functions such that*

$$|Q^{\pi_i}(s,a) - \tilde{Q}^{\pi_i}(s,a)| \leq \epsilon \text{ for all } s \in S, a \in A, \text{ and } i \in \{1,2,...,n\}.$$

*Define*

$$\pi(s) \in \operatorname*{argmax}_{a} \max_{i} \tilde{Q}^{\pi_i}(s,a).$$

*Then,*

$$Q^{\pi}(s,a) \geq \max_{i} Q^{\pi_i}(s,a) - \frac{2}{1-\gamma}\epsilon$$

*for any $s \in S$ and any $a \in A$, where $Q^{\pi}$ is the action-value function of $\pi$.*

*Proof.* To simplify the notation, let

$$Q_{\max}(s,a) = \max_{i} Q^{\pi_i}(s,a) \quad \text{and} \quad \tilde{Q}_{\max}(s,a) = \max_{i} \tilde{Q}^{\pi_i}(s,a).$$

We start by noting that for any $s \in S$ and any $a \in A$ the following holds:

$$|Q_{\max}(s,a) - \tilde{Q}_{\max}(s,a)| = |\max_{i} Q^{\pi_i}(s,a) - \max_{i} \tilde{Q}^{\pi_i}(s,a)| \leq \max_{i} |Q^{\pi_i}(s,a) - \tilde{Q}^{\pi_i}(s,a)| \leq \epsilon.$$

For all $s \in S$, $a \in A$, and $i \in \{1, 2, ..., n\}$ we have

$$
\begin{aligned}
T^\pi \tilde{Q}_{\max}(s, a) &= r(s, a) + \gamma \sum_{s'} p(s'|s, a) \tilde{Q}_{\max}(s', \pi(s')) \\
&= r(s, a) + \gamma \sum_{s'} p(s'|s, a) \max_b \tilde{Q}_{\max}(s', b) \\
&\geq r(s, a) + \gamma \sum_{s'} p(s'|s, a) \max_b Q_{\max}(s', b) - \gamma\epsilon \\
&\geq r(s, a) + \gamma \sum_{s'} p(s'|s, a) Q_{\max}(s', \pi_i(s')) - \gamma\epsilon \\
&\geq r(s, a) + \gamma \sum_{s'} p(s'|s, a) Q^{\pi_i}(s', \pi_i(s')) - \gamma\epsilon \\
&= T^{\pi_i} Q^{\pi_i}(s, a) - \gamma\epsilon \\
&= Q^{\pi_i}(s, a) - \gamma\epsilon.
\end{aligned}
$$

Since $T^\pi \tilde{Q}_{\max}(s, a) \geq Q^{\pi_i}(s, a) - \gamma\epsilon$ for any $i$, it must be the case that

$$
\begin{aligned}
T^\pi \tilde{Q}_{\max}(s, a) &\geq \max_i Q^{\pi_i}(s, a) - \gamma\epsilon \\
&= Q_{\max}(s, a) - \gamma\epsilon \\
&\geq \tilde{Q}_{\max}(s, a) - \epsilon - \gamma\epsilon.
\end{aligned}
$$

Let $e(s, a) = 1$ for all $s, a \in S \times A$. It is well known that $T^\pi(\tilde{Q}_{\max}(s, a) + ce(s, a)) = T^\pi \tilde{Q}_{\max}(s, a) + \gamma c$ for any $c \in \mathbb{R}$. Using this fact together with the monotonicity and contraction properties of the Bellman operator $T^\pi$, we have

$$
\begin{aligned}
Q^\pi(s, a) &= \lim_{k \to \infty} (T^\pi)^k \tilde{Q}_{\max}(s, a) \\
&\geq \tilde{Q}_{\max}(s, a) - \frac{1 + \gamma}{1 - \gamma} \epsilon \\
&\geq Q_{\max}(s, a) - \epsilon - \frac{1 + \gamma}{1 - \gamma} \epsilon.
\end{aligned}
$$

$\square$

**Lemma 1.** *Let $\delta_{ij} = \max_{s,a} |r_i(s, a) - r_j(s, a)|$. Then,*

$$
Q_i^{\pi_i^*}(s, a) - Q_i^{\pi_j^*}(s, a) \leq \frac{2\delta_{ij}}{1 - \gamma}.
$$

*Proof.* To simplify the notation, let $Q_i^j(s, a) \equiv Q_i^{\pi_j^*}(s, a)$. Then,

$$
\begin{aligned}
Q_i^i(s, a) - Q_i^j(s, a) &= Q_i^i(s, a) - Q_j^j(s, a) + Q_j^j(s, a) - Q_i^j(s, a) \\
&\leq |Q_i^i(s, a) - Q_j^j(s, a)| + |Q_j^j(s, a) - Q_i^j(s, a)|. \quad (11)
\end{aligned}
$$

Our strategy will be to bound $|Q_i^i(s, a) - Q_j^j(s, a)|$ and $|Q_j^j(s, a) - Q_i^j(s, a)|$. Note that $|Q_i^i(s, a) - Q_j^j(s, a)|$ is the difference between the value functions of two MDPs with the same transition function

but potentially different rewards. Let $\Delta_{ij} = \max_{s,a} |Q_i^i(s,a) - Q_j^j(s,a)|$. Then, [2]

$$
\begin{aligned}
|Q_i^i(s,a) - Q_j^j(s,a)| &= \left| r_i(s,a) + \gamma \sum_{s'} p(s'|s,a) \max_b Q_i^i(s',b) - r_j(s,a) - \gamma \sum_{s'} p(s'|s,a) \max_b Q_j^j(s',b) \right| \\
&= \left| r_i(s,a) - r_j(s,a) + \gamma \sum_{s'} p(s'|s,a) \left( \max_b Q_i^i(s',b) - \max_b Q_j^j(s',b) \right) \right| \\
&\leq |r_i(s,a) - r_j(s,a)| + \gamma \sum_{s'} p(s'|s,a) \left| \max_b Q_i^i(s',b) - \max_b Q_j^j(s',b) \right| \\
&\leq |r_i(s,a) - r_j(s,a)| + \gamma \sum_{s'} p(s'|s,a) \max_b \left| Q_i^i(s',b) - Q_j^j(s',b) \right| \\
&\leq \delta_{ij} + \gamma \Delta_{ij}.
\end{aligned}
\tag{12}
$$

Since (12) is valid for any $s, a \in S \times A$, we have shown that $\Delta_{ij} \leq \delta_{ij} + \gamma \Delta_{ij}$. Solving for $\Delta_{ij}$ we get

$$
\Delta_{ij} \leq \frac{1}{1-\gamma} \delta_{ij}.
\tag{13}
$$

We now turn our attention to $|Q_j^j(s,a) - Q_i^j(s,a)|$. Following the previous steps, define $\Delta'_{ij} = \max_{s,a} |Q_i^j(s,a) - Q_i^j(s,a)|$. Then,

$$
\begin{aligned}
|Q_j^j(s,a) - Q_i^j(s,a)| &= \left| r_j(s,a) + \gamma \sum_{s'} p(s'|s,a) Q_j^j(s', \pi_j^*(s')) - r_i(s,a) - \gamma \sum_{s'} p(s'|s,a) Q_i^j(s', \pi_j^*(s')) \right| \\
&= \left| r_i(s,a) - r_j(s,a) + \gamma \sum_{s'} p(s'|s,a) \left( Q_j^j(s', \pi_j^*(s')) - Q_i^j(s', \pi_j^*(s')) \right) \right| \\
&\leq |r_i(s,a) - r_j(s,a)| + \gamma \sum_{s'} p(s'|s,a) \left| Q_j^j(s', \pi_j^*(s')) - Q_i^j(s', \pi_j^*(s')) \right| \\
&\leq \delta_{ij} + \gamma \Delta'_{ij}.
\end{aligned}
$$

Solving for $\Delta'_{ij}$, as above, we get

$$
\Delta'_{ij} \leq \frac{1}{1-\gamma} \delta_{ij}.
\tag{14}
$$

Plugging (13) and (14) back in (11) we get the desired result. □

**Theorem 2.** *Let $M_i \in \mathcal{M}^\phi$ and let $Q_i^{\pi_j^*}$ be the value function of an optimal policy of $M_j \in \mathcal{M}^\phi$ when executed in $M_i$. Given the set $\{\tilde{Q}_i^{\pi_1^*}, \tilde{Q}_i^{\pi_2^*}, ..., \tilde{Q}_i^{\pi_n^*}\}$ such that*

$$
\left| Q_i^{\pi_j^*}(s,a) - \tilde{Q}_i^{\pi_j^*}(s,a) \right| \leq \epsilon \text{ for all } s \in S, a \in A, \text{ and } j \in \{1, 2, ..., n\},
$$

*let*

$$
\pi(s) \in \operatorname*{argmax}_a \max_j \tilde{Q}_i^{\pi_j^*}(s,a).
$$

*Finally, let $\phi_{\max} = \max_{s,a} ||\phi(s,a)||$, where $|| \cdot ||$ is the norm induced by the inner product adopted. Then,*

$$
Q_i^*(s,a) - Q_i^\pi(s,a) \leq \frac{2}{1-\gamma} \left( \phi_{\max} \min_j ||\mathbf{w}_i - \mathbf{w}_j|| + \epsilon \right).
$$

*Proof.* The result is a direct application of Theorem 1 and Lemma 1. For any $j \in \{1, 2, ..., n\}$, we have

$$
\begin{aligned}
Q_i^*(s,a) - Q_i^\pi(s,a) \quad &\leq Q_i^*(s,a) - Q_i^{\pi_j^*}(s,a) + \frac{2}{1-\gamma}\epsilon && \text{(Theorem 1)} \\
&\leq \frac{2}{1-\gamma} \max_{s,a} |r_i(s,a) - r_j(s,a)| + \frac{2}{1-\gamma}\epsilon && \text{(Lemma 1)} \\
&= \frac{2}{1-\gamma} \max_{s,a} |\phi(s,a)^\top \mathbf{w}_i - \phi(s,a)^\top \mathbf{w}_j| + \frac{2}{1-\gamma}\epsilon \\
&= \frac{2}{1-\gamma} \max_{s,a} |\phi(s,a)^\top (\mathbf{w}_i - \mathbf{w}_j)| + \frac{2}{1-\gamma}\epsilon \\
&\leq \frac{2}{1-\gamma} \max_{s,a} ||\phi(s,a)|| \, ||\mathbf{w}_i - \mathbf{w}_j|| + \frac{2}{1-\gamma}\epsilon && \text{(Cauchy-Schwarz's inequality)} \\
&= \frac{2\phi_{\max}}{1-\gamma} ||\mathbf{w}_i - \mathbf{w}_j|| + \frac{2}{1-\gamma}\epsilon.
\end{aligned}
$$

$\square$

# B  Details of the experiments

In this section we provide additional information about our experiments. We start with the four-room environment and then we discuss the reacher domain. In both cases the structure of the discussion is the same: we start by giving a more in depth description of the environment itself, both at a conceptual level and at a practical level, then we provide a thorough description of the algorithms used, and, finally, we explain the protocol used to carry out the experiments.

## B.1  Four-room environment

### B.1.1  Environment

In Section 5 of the paper we gave an intuitive description of the four-room domain used in our experiments. In this section we provide a more formal definition of the environment $\mathcal{M}$ as a family of Markov decision processes (MDPs) $M$, each one associated with a task.

The environment has objects that can be picked up by the agent by passing over them. There is a total of $n_o$ objects, each belonging to one of $n_c \leq n_o$ classes. The class of an object determines the reward $r_c$ associated with it. An episode ends when the agent reaches the goal, upon which all the objects re-appear. We assume that $r_g$ is always 1 but $r_c$ may vary: a specific instantiation of the rewards $r_c$ defines a *task*. Every time a new task starts the rewards $r_c$ are sampled from a uniform distribution over $[-1, 1]$. Figure 1 shows the specific environment layout used, in which $n_o = 12$ and $n_c = 3$.

We now focus on a single task $M \in \mathcal{M}$. We start by describing the state and action spaces, and the transition dynamics. The agent's position at any instant in time is a point $\{s_x, s_y\} \in [0,1]^2$. There are four actions available, $\mathcal{A} \equiv \{\text{up}, \text{down}, \text{left}, \text{right}\}$. The execution of one of the actions moves the agent 0.05 units in the desired direction, and normal random noise with zero mean and standard deviation 0.005 is added to the position of the agent (that is, a move along the $x$ axis would be $s_x' = s_x \pm \mathcal{N}(0.05, 0.005)$, where $\mathcal{N}(0.05, 0.005)$ is a normal variable with mean 0.05 and standard deviation 0.005). If after a move the agent ends up outside of the four rooms or on top of a wall the move is undone. Otherwise, if the agent lands on top of an object it picks it up, and if it lands on the goal region the episode is terminated (and all objects re-appear). In the specific instance of the environment shown in Figure 1 objects were implemented as circles of radius 0.04, the goal is a circle of radius 0.1 centered at one of the extreme points of the map, and the walls are rectangles of width 0.04 traversing the environment's range.

As described in the paper, within an episode each of the $n_o$ objects can be present or absent, and since they define the reward function a well defined Markov state must distinguish between all possible $2^{n_o}$ object configurations. Therefore, the state space of our MDP is $\mathcal{S} \equiv \{0,1\}^{n_o} \times \mathbb{R}^2$. An intuitive way of visualizing $\mathcal{S}$ is to note that each of the $2^{n_o}$ object configurations is potentially associated with a different value function over the continuous space $[0,1]^2$.

Having already described $\mathcal{S}$, $\mathcal{A}$, and $p(\cdot|s,a)$, we only need to define the reward function $R(s,a,s')$ and the discount factor $\gamma$ in order to conclude the formulation of the MDP $M$. As discussed in

Section 5, the reward $R(s, a, s')$ is a deterministic function of $s'$: if the agent is over an object of class $c$ in $s'$ it gets a reward of $r_c$, and if it is in the goal region it gets a reward of $r_g = 1$; in all other cases the reward is zero. In our experiments we fixed $\gamma = 0.95$.

By mapping each object onto its class, it is possible to construct features $\phi(s, a, s')$ that perfectly predicts the reward for all tasks $M$ in the environment $\mathcal{M}^\phi$, as in (2) and (5). Let $\phi_c(s, a, s') \equiv \delta\{$is the agent over an object of class $c$ in $s'$?$\}$, where $\delta\{\text{false}\} = 0$ and $\delta\{\text{true}\} = 1$. Similarly, let $\phi_g(s, a, s') \equiv \delta\{$is the agent over the goal region in $s'$?$\}$. By concatenating the $n_c$ functions $\phi_c$ and $\phi_g$ we get the vector $\phi(s, a, s') \in \{0, 1\}^{n_c+1}$; now, if we make $\mathbf{w}_c = r_c$ for all $c$ and $\mathbf{w}_{n_c+1} = r_g$, it should be clear that $r(s, a, s') = \phi(s, a, s')^\top \mathbf{w}$, as desired.

Since $r(s, a, s')$ can be written in the form of (2), the definition of $M$ can be naturally extended to $\mathcal{M}$, as in (5). In our experiments we assume that the agents receive a signal from $\mathcal{M}$ whenever the task changes (see Algorithms 1, 2, and 3 and discussion below).

### B.1.2 Algorithms

We assume that the agents know their position $\{s_x, s_y\} \in [0, 1]^2$ and also have an "object detector" $\mathbf{o} \in \{0, 1\}^{n_o}$ whose i-th component is 1 if and only if the agent is over object $i$. Using this information the agents build two vectors of features. The vector $\varphi_p(s) \in \mathbb{R}^{100}$ is composed of the activations of a regular $10 \times 10$ grid of radial basis functions at the point $\{s_x, s_y\}$. Specifically, in our experiments we used Gaussian functions, that is:

$$\varphi_{pi}(s) = \exp\left(-\frac{(s_x - \mathbf{c}_{i1})^2 + (s_y - \mathbf{c}_{i2})^2}{\sigma}\right), \tag{15}$$

where $\mathbf{c}_i \in \mathbb{R}^2$ is the center of the i-th Gaussian. As explained in Section B.1.3, the value of $\sigma$ was determined in preliminary experiments with QL; all algorithms used $\sigma = 0.1$. In addition to $\varphi_p(s)$, using $\mathbf{o}$ the agents build an "inventory" $\varphi_i(s) \in \{0, 1\}^{n_o}$ whose i-th component indicates whether the i-th object has been picked up or not. The concatenation of $\varphi_i(s)$ and $\varphi_p(s)$ plus a constant term gives rise to the feature vector $\varphi(s) \in \mathbb{R}^D$ used by all the agents to represent the value function: $\tilde{Q}^\pi(s, a) = \varphi(s)^\top \mathbf{z}_a^\pi$, where $\mathbf{z}_a^\pi \in \mathbb{R}^D$ are learned weights.

It is instructive to take a closer look at how exactly SFQL represents the value function. Note that, even though our algorithm also represents $\tilde{Q}^\pi$ as a linear combination of the features $\varphi(s)$, it never explicitly computes $\mathbf{z}_a^\pi$. Specifically, SFQL represent SFs as $\tilde{\psi}^\pi(s, a) = \varphi(s)^\top \mathbf{Z}_a^\pi$, where $\mathbf{Z}_a^\pi \in \mathbb{R}^{D \times d}$, and the value function as $\tilde{Q}^\pi(s, a) = \tilde{\psi}^\pi(s, a)^\top \tilde{\mathbf{w}} = \varphi(s)^\top \mathbf{Z}_a^\pi \tilde{\mathbf{w}}$. By making $\mathbf{z}_a^\pi = \mathbf{Z}_a^\pi \tilde{\mathbf{w}}$, it becomes clear that SFQL unfolds the problem of learning $\mathbf{z}_a^\pi$ into the sub-problems of learning $\mathbf{Z}_a^\pi$ and $\tilde{\mathbf{w}}$. These parameters are learned via gradient descent in order to minimize losses induced by (4) and (2), respectively (see below).

The pseudo-codes of QL, PRQL, and SFQL are given in Algorithms 1, 2, and 3, respectively. As one can see, all algorithms used an $\epsilon$-greedy policy to explore the environment, with $\epsilon = 0.15$ [21]. Two design choices deserve to be discussed here. First, as mentioned in Section B.1.1, the agents "know" when the task changes. This makes it possible for the algorithms to take measures like reinitializing the weights $\mathbf{z}_a^\pi$ or adding a new representative to the set of decision policies. Another design choice, this one specific to PRQL and SFQL, was not to limit the number of decision policies (or $\tilde{\psi}^{\pi_i}$) stored. It is not difficult to come up with strategies to avoid both an explicit end-of-task signal and an ever-growing set of policies. For example, in Section 4.2 we discuss how $\tilde{\mathbf{w}}$ can be used to select which $\tilde{\psi}^{\pi_i}$ to keep in the case of limited memory. Also, by monitoring the error in the approximation $\tilde{\phi}(s, a, s')^\top \tilde{\mathbf{w}}$ one can detect when the task has changed in a significant way. Although these are interesting extensions, given the introductory character of this paper we refrained from overspecializing the algorithms in order to illustrate the properties of the proposed approach in the clearest way possible.

### SFQL

We now discuss some characteristics of the specific SFQL algorithm used in the experiments. First, note that errors in the value-function approximation can potentially have a negative effect on GPI, since an overestimated $\tilde{Q}^{\pi_i}(s, a)$ may be the function determining the action selected by $\pi$ in (7). One

**Algorithm 1** QL

**Require:** $\epsilon$    exploration parameter for $\epsilon$-greedy strategy
            $\alpha$    learning rate
1: **for** $t \leftarrow 1, 2, ..., \text{num\_tasks}$ **do**
2:    $\mathbf{z}_a \leftarrow$ small random initial values **for all** $a \in \mathcal{A}$
3:    new\_episode $\leftarrow$ true
4:    **for** $i \leftarrow 1, 2, ..., \text{num\_steps}$ **do**
5:      **if** new\_episode **then**
6:        new\_episode $\leftarrow$ false
7:        $s \leftarrow$ initial state
8:      **end if**
9:      sel\_rand\_a $\sim$ Bernoulli($\epsilon$)          // Sample from a Bernoulli distribution with parameter $\epsilon$
10:      **if** sel\_rand\_a **then** $a \sim \text{Uniform}(\{1, 2, ..., |A|\})$       // $\epsilon$-greedy exploration strategy
11:      **else** $a \leftarrow \text{argmax}_b Q(s, b)$
12:      Take action $a$ and observe reward $r$ and next state $s'$
13:      **if** $s'$ is a terminal state **then**
14:        $\gamma \leftarrow 0$
15:        new\_episode $\leftarrow$ true
16:      **end if**
17:      $\mathbf{z}_a \leftarrow \mathbf{z}_a + \alpha(r + \gamma \max_{a'} Q(s', a') - Q(s, a)) \nabla_{\mathbf{z}} Q(s, a)$
                                   // For $Q(s, a) = \boldsymbol{\varphi}(s)^\top \mathbf{z}_a$, $\nabla_{\mathbf{z}} Q(s, a) = \boldsymbol{\varphi}(s)$
18:      $s \leftarrow s'$
19:    **end for**
20: **end for**

way of keeping this phenomenon from occurring indefinitely is to continue to update the functions $\tilde{Q}^{\pi_i}(s, a)$ that are relevant for action selection. In the context of SFs this corresponds to constantly refining $\tilde{\boldsymbol{\psi}}^{\pi_i}$, which can be done as long as we have access to $\pi_i$. In the scenario considered here we can recover $\pi_i$ by keeping the weights $\tilde{\mathbf{w}}_i$ used to learn the SFs $\tilde{\boldsymbol{\psi}}^{\pi_i}$ (line 2 of Algorithm 3). Note that with this information one can easily update any $\tilde{\boldsymbol{\psi}}^{\pi_i}$ off-policy; as shown in lines 25–30 of Algorithm 3, in the version of SFQL used in the experiments we always update the $\tilde{\boldsymbol{\psi}}^{\pi_i}$ that achieves the maximum in (7) (line 10 of the pseudo-code).

Next we discuss the details of how $\tilde{\mathbf{w}}$ and $\tilde{\boldsymbol{\psi}}^\pi$ are learned. We start by showing the loss function used to compute $\tilde{\mathbf{w}}$:

$$\mathrm{L}_w(\tilde{\mathbf{w}}) = \mathrm{E}_{(s,a,s') \sim \mathcal{D}} \left[ \left( r(s, a, s') - \tilde{\boldsymbol{\phi}}(s, a, s')^\top \tilde{\mathbf{w}} \right)^2 \right], \tag{16}$$

where $\mathcal{D}$ is a distribution over $\mathcal{S} \times \mathcal{A} \times \mathcal{S}$ which in RL is usually the result of executing a policy under the environment's dynamics $p(\cdot|s, a)$. The minimization of (16) is done in line 21 of Algorithm 3. As discussed, SFQL keeps a set of $\tilde{\boldsymbol{\psi}}^{\pi_i}$, each one associated with a policy $\pi_i$. The loss function used to compute each $\tilde{\boldsymbol{\psi}}^{\pi_i}$ is

$$
\begin{aligned}
\mathrm{L}_Z(\tilde{\boldsymbol{\psi}}^{\pi_i}) \equiv \mathrm{L}_Z(\mathbf{Z}_a^{\pi_i}) \quad &= \mathrm{E}_{(s,a,s') \sim \mathcal{D}} \left[ \left( \tilde{\boldsymbol{\phi}}(s, a, s') + \gamma \tilde{\boldsymbol{\psi}}^{\pi_i}(s', a') - \tilde{\boldsymbol{\psi}}^{\pi_i}(s, a) \right)^2 \right] \\
&= \mathrm{E}_{(s,a,s') \sim \mathcal{D}} \left[ \left( \tilde{\boldsymbol{\phi}}(s, a, s') + \gamma \boldsymbol{\varphi}(s')^\top \mathbf{Z}_{a'}^{\pi_i} - \boldsymbol{\varphi}(s)^\top \mathbf{Z}_a^{\pi_i} \right)^2 \right],
\end{aligned}
\tag{17}
$$

where $a' = \text{argmax}_b \tilde{Q}^{\pi_i}(s', b) = \text{argmax}_b \tilde{\boldsymbol{\psi}}^{\pi_i}(s', b)^\top \tilde{\mathbf{w}}$. Note that the policy that induces $\mathcal{D}$ is not necessarily $\pi_i$—that is, $\tilde{\boldsymbol{\psi}}^{\pi_i}(s, a)$ can be learned off-policy, as discussed above. As usual in RL, the target $\tilde{\boldsymbol{\phi}}(s, a, s') + \gamma \boldsymbol{\varphi}(s')^\top \mathbf{Z}_{a'}^{\pi_i}$ is considered fixed, *i.e.*, the loss $\mathrm{L}_Z$ is minimized with respect to $\mathbf{Z}_a^{\pi_i}$ only. The minimization of (17) is done in lines 23 and 28 of Algorithm 3.

As discussed in Section 5, we used two versions of SFQL in our experiments. In the first one, SFQL-$\phi$, we assume that the agent knows how to construct a vector of features $\phi$ that perfectly predicts the reward for all tasks $M$ in the environment $\mathcal{M}^\phi$. The other version of our algorithm, SFQL-$h$, uses an approximate $\tilde{\phi}$ learned from data. We now give details of how $\tilde{\phi}$ was computed

---

**Algorithm 2** PRQL

---

**Require:**
$\epsilon$    exploration parameter for $\epsilon$-greedy strategy
$\alpha$    learning rate
$\eta$    parameter to control probability to reuse old policy
$\tau$    parameter to control bias in favor of stronger policies

1: **for** $t \leftarrow 1, 2, ..., \text{num\_tasks}$ **do**
2:    **for** $k \leftarrow 1, 2, ..., t$ **do**
3:      $\text{score}_k \leftarrow 0$                  // $\text{score}_k$ is the score associated with policy $\pi_k$
4:      $\text{used}_k \leftarrow 0$                  // $\text{used}_k$ is the number of times policy $\pi_k$ was used
5:    **end for**
6:    $c \leftarrow t$                         // c is the index of the policy currently being used
7:    $\mathbf{z}_a^t \leftarrow$ small random initial values
8:    $\text{curr\_score} \leftarrow 0$
9:    $\text{new\_episode} \leftarrow$ true
10:    **for** $i \leftarrow 1, 2, ..., \text{num\_steps}$ **do**
11:      **if** new\_episode **then**
12:        $\text{score}_c \leftarrow \dfrac{\text{score}_c \times \text{used}_c + \text{curr\_score}}{\text{used}_c + 1}$      // Update score for policy currently being used
13:        **for** $k \leftarrow 1, 2, ..., t$ **do** $p_k \leftarrow e^{\tau \times \text{score}_k} / \sum_{j=1}^{t} e^{\tau \times \text{score}_j}$    // Turn scores into probabilities
14:        $c \sim \text{Multinomial}(p_1, p_2, ..., p_t)$          // Select policy c with probability $p_c$
15:        $\text{used}_c \leftarrow \text{used}_c + 1$          // Update number of times policy $\pi_c$ has been used
16:        $\text{curr\_score} \leftarrow 0$
17:        $\text{new\_episode} \leftarrow$ false
18:        $s \leftarrow$ initial state
19:      **end if**
20:      **if** $t \neq c$ **then** use\_prev\_policy $\sim \text{Bernoulli}(\eta)$ **else** use\_prev\_policy $\leftarrow$ false
21:      **if** use\_prev\_policy **then**         // Action will be selected by $\pi_c$, the policy being reused
22:        $a \leftarrow \text{argmax}_{a'} Q_c(s, a')$
23:      **else**                 // Action will be selected by $\pi_c$, the most recent policy
24:        $\text{sel\_rand\_a} \sim \text{Bernoulli}(\epsilon)$
25:        **if** sel\_rand\_a **then** $a \sim \text{Uniform}(\{1, 2, ..., |A|\})$ **else** $a \leftarrow \text{argmax}_{a'} Q_t(s, a')$
                                              // $\epsilon$-greedy exploration strategy
26:      **end if**
27:      Take action $a$ and observe reward $r$ and next state $s'$
28:      **if** $s'$ is a terminal state **then**
29:        $\gamma \leftarrow 0$
30:        $\text{new\_episode} \leftarrow$ true
31:      **end if**
32:      $\mathbf{z}_a^t \leftarrow \mathbf{z}_a^t + \alpha(r + \gamma \max_{a'} Q_t(s', a') - Q_t(s, a)) \nabla_{\mathbf{z}} Q_t(s, a)$
                                // For $Q_t(s, a) = \boldsymbol{\varphi}(s)^{\top} \mathbf{z}_a^t$, $\nabla_{\mathbf{z}} Q_t(s, a) = \boldsymbol{\varphi}(s)$
33:      $\text{curr\_score} \leftarrow \text{curr\_score} + r$
34:      $s \leftarrow s'$
35:    **end for**
36: **end for**

---

in this case. In order to learn $\tilde{\phi}$, we used the samples $(s_i, a_i, r_i, s_i')_t$ collected by QL in the first $t = 1, 2, ..., 20$ tasks. Since this results in an unbalanced dataset in which most of the transitions have $r_i = 0$, we kept all the samples with $r_i \neq 0$ and discarded 75% of the remaining samples. We then used the resulting dataset to minimize the following loss:

$$\text{L}_H(\tilde{\phi}) \equiv \text{L}_H(\mathbf{H}, \mathbf{w}_t) = \text{E}_{(s,s',r) \sim \mathcal{D}_t'} \left[ \left( \varsigma(\boldsymbol{\varphi}(s, s')^{\top} \mathbf{H})^{\top} \mathbf{w}_t - r \right)^2 \right] \quad \text{for } t = 1, 2, ..., 20, \quad (18)$$

where $\mathcal{D}_t'$ reflects the down-sampling of zero rewards. The vector of features $\boldsymbol{\varphi}(s, s')$ is the concatenation of $\boldsymbol{\varphi}(s)$ and $\boldsymbol{\varphi}(s')$, and $\varsigma(\cdot)$ is a sigmoid function applied element-wise. We note that $\mathbf{o}(s') = \boldsymbol{\varphi}_i(s') - \boldsymbol{\varphi}_i(s)$, from which it is possible to compute an "exact" $\tilde{\phi} = \phi$. In order to minimize (18) we used the multi-task framework proposed by Caruana [6]. Simply put, Caruana's [6] approach consists in looking at $\varsigma(\boldsymbol{\varphi}(s, s')^{\top} \mathbf{H})^{\top} \mathbf{w}_t$ as a neural network with one hidden layer and 20

---
**Algorithm 3** SFQL
---

**Require:**
    $\epsilon$    exploration parameter for $\epsilon$-greedy strategy
    $\alpha$    learning rate for $\psi$'s parameters
    $\alpha_w$    learning rate for $\mathbf{w}$
    $\phi$    features to be predicted by SFs

1: **for** $t \leftarrow 1, 2, ..., \text{num\_tasks}$ **do**
2:     $\mathbf{w}_t \leftarrow$ small random initial values
3:     $\mathbf{Z}_a^t \leftarrow$ small random initial values in $\mathbb{R}^{D \times h}$ **if** $t = 1$ **else** $\mathbf{Z}_a^{t-1}$
         // The $\text{k}^{\text{th}}$ column of $\mathbf{Z}_a^t$, $\mathbf{z}_a^{tk}$, are the parameters of the $\text{k}^{\text{th}}$ component of $\tilde{\psi}_t$, $(\tilde{\psi}_t)_k \equiv \tilde{\psi}_{tk}$
4:     $\text{new\_episode} \leftarrow \text{true}$
5:     **for** $i \leftarrow 1, 2, ..., \text{num\_steps}$ **do**
6:         **if** $\text{new\_episode}$ **then**
7:             $\text{new\_episode} \leftarrow \text{false}$
8:             $s \leftarrow$ initial state
9:         **end if**
10:        $c \leftarrow \text{argmax}_{k \in \{1,2,...,t\}} \max_b \tilde{\psi}_k(s, b)^\top \mathbf{w}_t$
          // $c$ is the index of the $\tilde{\psi}$ associated with the largest value in $s$
11:        $\text{sel\_rand\_a} \sim \text{Bernoulli}(\epsilon)$      // Sample from a Bernoulli distribution with parameter $\epsilon$
12:        **if** $\text{sel\_rand\_a}$ **then** $a \sim \text{Uniform}(\{1, 2, ..., |A|\})$      // $\epsilon$-greedy exploration strategy
13:        **else** $a \leftarrow \text{argmax}_b \tilde{\psi}_c(s, b)^\top \mathbf{w}_t$
14:        Take action $a$ and observe reward $r$ and next state $s'$
15:        **if** $s'$ is a terminal state **then**
16:            $\gamma \leftarrow 0$
17:            $\text{new\_episode} \leftarrow \text{true}$
18:        **else**
19:            $a' \leftarrow \text{argmax}_b \max_{k \in \{1,2,...,t\}} \tilde{\psi}_k(s', b)^\top \mathbf{w}_t$   // $a'$ is the action with the highest value in $s'$
20:        **end if**
21:        $\mathbf{w}_t \leftarrow \mathbf{w}_t + \alpha_w \big[r - \phi(s, a, s')^\top \mathbf{w}\big] \phi(s, a, s')$         // Update $\mathbf{w}$
22:        **for** $k \leftarrow 1, 2, ..., d$ **do**
23:            $\mathbf{z}_a^{tk} \leftarrow \mathbf{z}_a^{tk} + \alpha \Big[\phi_k(s, a, s') + \gamma \tilde{\psi}_{tk}(s', a') - \tilde{\psi}_{tk}(s, a)\Big] \nabla_{\mathbf{z}} \tilde{\psi}_{tk}(s, a)$
          // For $\tilde{\psi}_t(s, a) = \varphi(s)^\top \mathbf{Z}_a^t$, $\nabla_{\mathbf{z}} \tilde{\psi}_{tk}(s, a) = \varphi(s)$
24:        **end for**
25:        **if** $c \neq t$ **then**
26:            $a' \leftarrow \text{argmax}_b \tilde{\psi}_c(s', b)^\top \mathbf{w}_c$   // $a'$ is selected according to reward function induced by $\mathbf{w}_c$
27:            **for** $k \leftarrow 1, 2, ..., d$ **do**
28:                $\mathbf{z}_a^{ck} \leftarrow \mathbf{z}_a^{ck} + \alpha \Big[\phi_k(s, a, s') + \gamma \tilde{\psi}_{ck}(s', a') - \tilde{\psi}_{ck}(s, a)\Big] \nabla_{\mathbf{z}} \tilde{\psi}_{ck}(s, a)$   // Update $\tilde{\psi}_c$
29:            **end for**
30:        **end if**
31:        $s \leftarrow s'$
32:     **end for**
33: **end for**

---

outputs, that is, $\tilde{\mathbf{w}}$ is replaced with $\tilde{\mathbf{W}} \in \mathbb{R}^{h \times 20}$ and a reward $r$ received in the $\text{t}^{\text{th}}$ task is extended into a 20-dimensional vector in which the $\text{t}^{\text{th}}$ component is $r$ and all other components are zero. One can then minimize (18) with respect to the parameters $\mathbf{H}$ and $\mathbf{w}_t$ through gradient descent.

Although this strategy of using the $k$ first tasks to learn $\tilde{\phi}$ is feasible, in practice one may want to replace this arbitrary decision with a more adaptive approach, such as updating $\tilde{\phi}$ online until a certain stop criterion is satisfied. Note though that a significant change in $\tilde{\phi}$ renders the SFs $\tilde{\psi}^{\pi_i}$ outdated, and thus the benefits of refining the former should be weighed against the overhead of constantly updating the latter, potentially off-policy.

As one can see in this section, we tried to keep the methods as simple as possible in order to not obfuscate the main message of the paper, which is not to propose any particular algorithm but rather to present a general framework for transfer based on the combination of SFs and GPI.

### B.1.3 Experimental setup

In this section we describe the precise protocol adopted to carry out our experiments. Our initial step was to use QL, the basic algorithm behind all three algorithms, to make some decisions that apply to all of them. First, in order to define the features $\varphi(s)$ used by the algorithms, we checked the performance of QL when using different configurations of the vector $\varphi_p(s)$ giving the position of the agent. Specifically, we tried two-dimensional grids of Gaussians with 5, 10, 15, and 20 functions per dimension. Since the improvement in QL's performance when using a number of functions larger than 10 was not very significant, we adopted a $10 \times 10$ grid of Gaussians in our experiments. We also varied the value of the parameter $\sigma$ appearing in (15) in the set $\{0.01, 0.1, 0.3\}$. Here the best performance of QL was obtained with $\sigma = 0.1$, which was then the value adopted throughout the experiments. The parameter $\epsilon$ used for $\epsilon$-greedy exploration was also set based on QL's performance. Specifically, we varied $\epsilon$ in the set $\{0.15, 0.2, 0.3\}$ and verified that the best results were obtained with $\epsilon = 0.15$ (we tried relatively large values for $\epsilon$ because of the non-stationarity of the underlying environment). Finally, we tested two variants of QL: one that resets the weights $\mathbf{z}_a^\pi$ every time a new task starts, as in line 2 of Algorithm 1, and one that keeps the old values. Since the performance of the former was significantly better, we adopted this strategy for all algorithms.

QL, PRQL, and SFQL depend on different sets of parameters, as shown in Algorithms 1, 2, and 3. In order to properly configure the algorithms we tried three different values for each parameter and checked the performance of the corresponding agents under each resulting configuration. Specifically, we tried the following sets of values for each parameter:

| Parameter | Algorithms | Values |
|---|---|---|
| $\alpha$ | QL, PRQL, SFQL | $\{0.01, 0.05, 0.1\}$ |
| $\alpha_w$ | SFQL | $\{0.01, 0.05, 0.1\}$ |
| $\eta$ | PRQL | $\{0.1, 0.3, 0.5\}$ |
| $\tau$ | PRQL | $\{1, 10, 100\}$ |

The cross-product of the values above resulted in 3 configurations of QL, 27 configurations of PRQL, and 9 configurations of SFQL. The results reported correspond to the best performance of each algorithm, that is, for each algorithm we picked the configuration that lead to the highest average return over all tasks.

## B.2 Reacher environment

### B.2.1 Environment

The reacher environment is a two-joint torque-controlled robotic arm simulated using the MuJoCo physics engine [26]. It is based on one of the domains used by Lillicrap et al. [12]. This is a particularly appropriate domain to illustrate our ideas because it is straightforward to define multiple tasks (goal locations) sharing the same dynamics.

The problem's state space $\mathcal{S} \subset \mathbb{R}^4$ is composed of the angle and angular velocities of the two joints. The two-dimensional continuous action space $\mathcal{A}$ was discretized using 3 values per dimension (maximum positive, maximum negative or zero torque for each actuator), resulting in a total of 9 discrete actions. We adopted a discount factor of $\gamma = 0.9$.

The reward received at each time step was $-\delta$, where $\delta$ is the Euclidean distance between the target position and the tip of the arm. The start state at each episode was defined as follows during training: the inner joint angle was sampled from an uniform distribution over $[0, 2\pi]$, the outer joint was sampled from an uniform distribution over $\{-\pi/2, \pi/2\}$, and both angular velocities were set to $0$ (during the evaluation phase two fixed start states were used—see below). We used a time step of 0.02s and episodes lasted for 10s (500 time steps). We defined 12 target locations, 4 of which we used for training and 8 were reserved for testing (see Figure 3).

### B.2.2 Algorithms

The baseline method used for comparisons in the reacher domain was the DQN algorithm by Mnih et al. [15]. In order to make it possible for DQN to generalize across tasks we provided the target locations as part of the state description. The action-value function $\tilde{Q}$ was represented by a multi-layer perceptron (MLP) with two hidden layers of 256 linear units followed by $\tanh$ non-linearities. The

output of the network was a vector in $\mathbb{R}^9$ with the estimated value associated with each action. The replay buffer adopted was large enough to retain all the transitions seen by the agent—that is, we never removed transitions from the buffer (this helps prevent DQN from "forgetting" previously seen tasks when learning new ones). Each value-function update used a mini-batch of 32 transitions sampled uniformly from the replay buffer, and the associated minimization (line 17 of Algorithm 1) was carried out using the Adam optimizer with a learning rate of $10^{-3}$ [30].

SFDQN is similar to SFQL, whose pseudo-code is shown in Algorithm 3, with a few modifications to make learning with nonlinear function approximators more stable. The vector of features $\phi \in \mathbb{R}^{12}$ used by SFDQN was composed of the negation of the distances to all target locations.[3] We used a separate MLP to represent each of the four $\tilde{\psi}_i$. The MLP architecture was the same as the one adopted with DQN, except that in this case the output of the network was a matrix $\tilde{\Psi}_i \in \mathbb{R}^{9 \times 12}$ representing $\tilde{\psi}_i(s,a) \in \mathbb{R}^{12}$ for each $a \in \mathcal{A}$. This means that the parameters $\mathbf{Z}_i$ in Algorithm 3 should now be interpreted as weights of a nonlinear neural network. The final output of the network was computed as $\tilde{\psi}_i^\top \mathbf{w}_t$, where $\mathbf{w}_t \in \mathbb{R}^{12}$ is an one-hot vector indicating the task $t$ currently active. Analogously to the the target locations given as inputs to DQN, here we assume that $\mathbf{w}_t$ is provided by the environment. Again making the connection with Algorithm 3, this means that line 21 would be skipped.

Following Mnih et al. [15], in order to make the training of the neural network more stable we updated $\tilde{\psi}$ using a target network. This corresponds to replacing (17) with the following loss:

$$ \mathrm{L}_Z(\tilde{\psi}^{\pi_i}) \equiv \mathrm{L}_Z(\mathbf{Z}^{\pi_i}) = \mathrm{E}_{(s,a,s') \sim \mathcal{D}} \left[ \left( \phi(s,a,s') + \gamma \tilde{\psi}_-^{\pi_i}(s',a') - \tilde{\psi}^{\pi_i}(s,a) \right)^2 \right], $$

where $\tilde{\psi}_-^{\pi_i}$ is a target network that remained fixed during updates and was periodically replaced by $\tilde{\psi}^{\pi_i}$ at every 1000 steps (the same configuration used for DQN). For each transition we updated all four MLPs $\tilde{\psi}_i$ in order to minimize losses derived from (4). As explained in Section 5, the policies $\pi_i(s)$ used in (4) were the GPI policies associated with each training task, $\pi_i(s) \in \mathrm{argmax}_a \max_j \tilde{\psi}_j(s,a)^\top \mathbf{w}_i$. An easy way to modify Algorithm 3) to reflect this update strategy would be to select the action $a'$ in line 26 using $\pi_i(s)$ and then repeat the block in lines 25–30 for all $\mathbf{w}_i$, with $i \in \{1,2,3,4\}$ and $i \neq t$, where $t$ is the current task.

### B.2.3 Experimental setup

The agents were trained for $200\,000$ transitions on each of the 4 training tasks. Data was collected using an $\epsilon$-greedy policy with $\epsilon = 0.1$ (for SFDQN, this corresponds to lines 12 and 13 of Algorithm 3). As is common in fixed episode-length control tasks, we excluded the terminal transitions during training to make the value of states independent of time, which corresponds to learning continuing policies [12].

During the entire learning process we monitored the performance of the agents on all 12 tasks when using an $\epsilon$-greedy policy with $\epsilon = 0.03$. The results shown in Figure 3 reflect the performance of this policy. Specifically, the return shown in the figure is the sum of rewards received by the 0.03-greedy policy over two episodes starting from fixed states. Since the maximum possible return varies across tasks, we normalized the returns per task based on the performance of standard DQN on separate experiments (this is true for both training and test tasks). Specifically, we carried out the normalization as follows. First, we ran DQN 30 times on each task and recorded the algorithm's performance before and after training. Let $\bar{G}_b$ and $\bar{G}_a$ be the average performance of DQN over the 30 runs before and after training, respectively. Then, if during our actual experiments DQN or SFDQN got a return of $G$, the normalized version of this metric was obtained as $G_n = (G - \bar{G}_b)/(\bar{G}_a - \bar{G}_b)$. These are the values shown in Figure 3. Visual inspection of videos extracted from the experiments with DQN alone suggests that the returns used for normalization were obtained by near-optimal policies that reach the targets almost directly.

Figure 4: Understanding how much each type of transfer promoted by SFs helps in performance. Results averaged over 30 runs; standard errors are shown as shadowed regions but are almost imperceptible at this scale. PRQL's results are shown for reference.

## C   Additional empirical analysis

In this section we report empirical results that had to be left out of the main paper due to the space limit. Specifically, the objective of the experiments described here is to provide a deeper understanding of SFQL, in particular, and of SFs, more generally. We use the four-room environment to carry out our empirical investigation.

### C.1   Understanding the types of transfer promoted by SFs

We start by asking why exactly SFQL performed so much better than QL and PRQL in our experiments (see Figure 2). Note that there are several possible reasons for that to be the case. As shown in (3), SFQL uses a decoupled representation of the value function in which the environment's dynamics are dissociated from the rewards. If the reward function of a given environment can be non-trivially decomposed in the form (2) assumed by SFQL, the algorithm can potentially build on this knowledge to quickly learn the value function and to adapt to changes in the environment, as discussed in Section 3. In our experiments not only did we know that such a non-trivial decomposition exists, we actually provided such an information to SFQL— either directly, through a handcrafted $\phi$, or indirectly, by providing features that allow for $\phi$ to be recovered exactly. Although this fact should help explain the good results of SFQL, it does not seem to be the main reason for the difference in performance. Observe in Figure 2 how the advantage of SFQL over the other methods only starts to show up in the second task, and it only becomes apparent from the third task on. This suggests that the the algorithm's good performance is indeed due to some form of transfer.

But what kind of transfer, exactly? We note that SFQL naturally promotes two forms of transfer. The first one is of course a consequence of GPI. As discussed in the paper, SFQL applies GPI by storing a set of SFs $\tilde{\psi}^{\pi_i}$. Note though that SFs promote a weaker form of transfer even when only a *single* $\tilde{\psi}^{\pi}$ exists. To see why this is so, observe that if $\tilde{\psi}^{\pi}$ persists from task $t$ to $t + 1$, instead of arbitrary approximations $\tilde{Q}^{\pi}$ one will have reasonable estimates of $\pi$'s value function under the current $\tilde{\mathbf{w}}$. In other words, $\tilde{\psi}^{\pi}$ will transfer knowledge about $\pi$ from one task to the other.

In order to have a better understanding of the two types of transfer promoted by SFs, we carried out experiments in which we tried to isolate as much as possible each one of them. Specifically, we repeated the experiments shown in Figure 2 but now running SFQL with and without GPI (we can turn off GPI by replacing $c$ with $t$ in line 10 of Algorithm 3).

The results of our experiments are shown in Figure 4. It is interesting to note that even without GPI SFQL initially outperforms PRQL. This is probably the combined effect of the two factors discussed above: the decoupled representation of the value function plus the weak transfer promoted by $\tilde{\psi}^{\pi}$. Note though that, although these factors do give SFQL a head-start, eventually both algorithms reach the same performance level, as clear by the slope of the respective curves. In contrast, SFQL with GPI consistently outperforms the other two methods, which is evidence in favor of the hypothesis that GPI is indeed a crucial component of the proposed approach. Another evidence in this direction

$$\tilde{\boldsymbol{\phi}}(s,a,s')^{\top}\tilde{\mathbf{w}}_t \qquad \max_{a,i<t}\tilde{\boldsymbol{\psi}}_i(s,a)^{\top}\tilde{\mathbf{w}}_t \qquad \max_{a,i\leq t}\tilde{\boldsymbol{\psi}}_i(s,a)^{\top}\tilde{\mathbf{w}}_t$$

Figure 5: Functions computed by SFQL after 200 transitions into three randomly selected tasks (all objects present).

is given in Figure 5, which shows all the functions computed by the SFQL agent. Note how after only 200 transitions into a new task SFQL already has a good approximation of the reward function, which, combined with the set of previously computed $\tilde{\boldsymbol{\psi}}^{\pi_i}$, with $i < t$, provide a very informative value function even without the current $\tilde{\boldsymbol{\psi}}^{\pi_t}$.

## C.2 Analyzing the robustness of SFs

As discussed in the previous section, part of the benefits provided by SFs come from the decoupled representation of the value function shown in (3), which depends crucially on a decomposition of the reward function $\tilde{\boldsymbol{\phi}}(s,a,s')^{\top}\mathbf{w} \approx r(s,a,s')$. In Section 5 we illustrated two ways in which such a decomposition can be obtained: by handcrafting $\tilde{\boldsymbol{\phi}}$ based on prior knowledge about the environment or by learning it from data. When $\tilde{\boldsymbol{\phi}}$ is learned it is obviously not reasonable to expect an exact decomposition of the reward function, but even when it is handcrafted the resulting reward model can be only an approximation (for example, the "object detector" used by the agents in the experiments with the four-room environment could be noisy). Regardless of the source of the imprecision, we do not want SFQL in particular, and more generally any method using SFs, to completely break with the addition of noise to $\tilde{\boldsymbol{\phi}}$. In Section 5 we saw how an approximate $\tilde{\boldsymbol{\phi}}$ can in fact lead to very good performance. In this section we provide a more systematic investigation of this matter by analyzing the performance of SFQL with $\tilde{\boldsymbol{\phi}}$ corrupted in different ways.

Our first experiment consisted in adding spurious features to $\phi$ that do not help in any way in the approximation of the reward function. This reflects the scenario where the agent's set of predictions is a superset of the features needed to compute the reward (see in Section 6 the connection with Sutton et al.'s GVFs, 2011). In order to implement this, we added objects to our environment that always lead to zero reward—that is, even though the SFs $\tilde{\boldsymbol{\psi}}^{\pi}$ learned by SFQL would predict the occurrence of these objects, such predictions would not help in the approximation of the reward function. Specifically, we added to our basic layout 15 objects belonging to 6 classes, as shown in Figure 6, which means that in this case SFQL used a $\tilde{\boldsymbol{\phi}} \in \mathbb{R}^{10}$. In addition to adding spurious features, we also corrupted $\tilde{\boldsymbol{\phi}}$ by adding to each of its components, in each step, a sample from a normal distribution with mean zero and different standard deviations $\sigma$.

The outcome of our experiment is shown in Figure 7. Overall, the results are as expected, with both spurious features and noise hurting the performance of SFQL. Interestingly, the two types of perturbations do not seem to interact very strongly, since their effects seem to combine in an

Figure 6: Environment layout with additional objects. Spurious objects are represented as diamonds; numbers indicate the classes of the objects.

Figure 7: Analyzing SFQL's robustness to distortions in $\tilde{\phi}$. PRQL's and QL's results are shown for reference. Results averaged over 30 runs; standard errors are shown on top of each bar.

additive way. More important, SFQL's performance seems to degrade gracefully as a result of either intervention, which corroborate our previous experiments showing that the proposed approach is robust to approximation errors in $\tilde{\phi}$.

## Footnotes

[2] We follow the steps of Strehl and Littman [31].

[3]In fact, instead of the negation of the distances $-\delta$ we used $1 - \delta$ in the definition of both the rewards and the features $\phi_i$. Since in our domain $\delta < 1$, this change made the rewards always positive. This helps preventing randomly-initialized value function approximations from dominating the 'max' operation in (7).