[Reviews · NeurIPS 2017]

Reviewer 1



This paper combines ideas from Successor Representation and a generalization of the famous policy improvement step of Policy Iteration framework to propose a novel transfer learning algorithm, in the context of MDPs wherein the transition dynamics remain fixed but the reward function changes. The crux of the idea is a kind of two stage process: Stage1: A decoupled representation based model for the Q-function is used to learn successor features and the underlying reward vector for the MDP separately. Stage2: If transfer to task n+1 is to be shown, optimal successor features and resultant Q* functions are built for tasks 1...n. The resultant optimal Q* functions for task T_1.. T_n are used to perform a generalized form of policy improvement step in task n+1. Let the resultant policy be denoted by \pi. The authors demonstrate theoretical results for how close Q_{\pi} is to Q* for the n+1 task. The proposed method is empirically demonstrated on a gridworld task, as well as a reaching task. The use of successor representation for transfer is not novel, and this has appeared in various guises in the literature as pointed out in the nice lit review section. What appears to be novel is the particular representation mechanism chosen (SF) and the consequent applicability to continuous domains. The paper is very well written. The ideas are very clearly presented, and it was a pleasure to read. The empirical results were useful and the baselines used were reasonably powerful. The analysis is sufficiently detailed. The paper does make the point repeatedly, that this method is useful for building a library of skills. While one can see conceptually that this might be so, the experiments do not demonstrate this conclusively. There are many issues that need to be addressed as pointed out by the authors themselves to make this a functional option learning system.

Reviewer 2



This paper presents a RL optimization scheme and a theoretical analysis of its transfer performance. While the components of this work aren't novel, it combines them in an interesting, well-presented way that sheds new light. The definition of transfer given in Lines 89–91 is nonstandard. It seems to be missing the assumption that t is not in T. The role of T' is a bit strange, making this a requirement for "additional transfer" rather than just transfer. It should be better clarified that this is a stronger requirement than transfer, and explained what it's good for — the paper shows this stronger property holds, but never uses it. Among other prior work, Theorem 1 is closely related to Point-Based Value Iteration, and holds for the same reasons. In fact, w can be viewed as an unnormalized belief over which feature is important for the task (except that w can be negative). The corollary in Lines 163–165 can be strengthened by relaxing the condition to hold for *some* s, s' that have positive occurrences. The guarantee in (10) only comes from the closest w_j. Wouldn't it make sense to simplify and accelerate the algorithm by fixing this j at the beginning of the task? Or is there some benefit in practice beyond what the analysis shows? Empirical evidence either way would be helpful. Very much missing is a discussion of the limitations of this approach. For example, how badly does it fail when the feature-space dimension becomes large? Beyond the two 2D experiments presented, it's unclear how easy it is to find reward features such that similar tasks are close in feature-space. This seems to go to the heart of the challenge in RL, rather than to alleviate it. The connection drawn to temporal abstraction and options is intriguing but lacking. While \psi is clearly related to control and w to goals, the important aspect of identifying option termination is missing here.

Reviewer 3



This paper introduces a transfer learning method for MDPs with the same dynamics but different reward functions. It achieves this by using a representation that decouples rewards from MDP dynamics and combining this representation with a simple, but novel way of combining policies. The paper describes 2 main contributions: successor features and generalized policy improvement. Both ideas are extensions of existing work (successor states and policy improvement). Nonetheless, I feel both constitute a significant contribution. Successor features offer a principled way to generalize over families of tasks and their value functions. Though similar ideas have been explored in universal option models and UFVAs (as the authors also mention), the authors demonstrate that SF provide a very method for the transfer learning setting. The generalized policy improvement (Theorem 1) extends Bellman’s policy improvement and seems likely to find more applications in both multi-task and parallel learning settings. The paper combines both methods into an elegant approach to multi-task learning, which is supported with both theoretical and empirical results. The paper is very well written. It clearly introduces background and provides intuitive explanations as well as formal results for the presented ideas. The experiments provide thorough evaluations on a simple and a more involved testbed. The methods are compared both to a base learner and a simple transfer method. Overall, the experiments convincingly demonstrate the merit of the proposed method.